# Young People's Construction of Identity in the Context of Southern Europe: Finding Leads for Citizenship Education

Thiago Freires [1],*, Leanete Thomas Dotta [2] and Fátima Pereira [1]

1   Centre for Research and Intervention in Education (CIIE), Faculty of Psychology and Education Sciences, University of Porto, 4200-135 Porto, Portugal; fpereira@fpce.up.pt
2   Centre for Interdisciplinary Studies in Education and Development (CeiED), Lusófona University, 4000-098 Lisbon, Portugal; leanete.thomas@ulosofona.pt
*   Correspondence: tfreires@fpce.up.pt

**Abstract:** Identity building can be understood as a fluid process informed by sociocultural aspects and encompassing a strong dimension of othering. Relying on the notion of narrative identity, in this article, we explore the ways in which young people raise and discuss values (human dignity, freedom, solidarity, etc.). To do this, we draw on a set of data collected through deliberative discussions with 378 young people (11 to 20 years old) from four Southern European countries: Portugal, Spain, Italy, and Cyprus. These small group discussions confronted young people about their identification with their country and with Europe. Using thematic and descriptive analysis, we investigate the relationships established by young people with values spontaneously raised by them to build on identity formation. Our results reinforce identity as being constituted in varying forms across the European regions, with relation to values being plural. Yet, there is a strong reference to process values in the four participating countries, such as solidarity and equality, which seem to inform a narrative of an "inclusive Europe", where community ties matter. Because some level of controversy about values is observed, however, we argue that it could constitute a valuable aspect to inform activities in the field of citizenship education.

**Keywords:** citizenship education; southern Europe; youth identities

## 1. Introduction

It is well established in the literature that individuals are not restricted to singular identities [1–6]. For that reason, the examination of identity construction as a process benefits from a multidisciplinary perspective that extrapolates the field of psychology and integrates the dimensions of cultural diversity and elements derived from common and shared projects. This necessarily implies dialogue with the psychological dimension since it plays a relevant role in self-conception [6]. In this sense, identity can be characterized by local, national, supranational, or transnational layers of attachment [2], while encompassing a firm dimension of othering, i.e., the individuals mobilize their repertoire of identities according to who they are interacting with [3]. As Ross [3] affirms, "The identity or group of identities selected for presentation is a response to the group(s) that constitute the audience, to the location of the encounter, and to the history and events that preceded it" (p. 287–288).

Framed by the sociological approach, identity refers to the dynamics of self-conception and recognition, dealing with the meanings and the social norms that shape and connect individual and social behavior [2]. Rooted in social constructivism, Ross [3] adds that it is only possible to develop our sense of self-identity through social processes since we define ourselves in relation to others, resulting either in a position of grouping or dissociation [2]. Needless to say, others will simultaneously define our identity on their terms, according to their perceptions and constructions of what they believe or assume our identity to be, in processes that might result in identities which are different from those we would like to assume [3,7,8]. Identity development processes are crucially developed through the period

of young adulthood, since this comprises a moment when individuals try to explore and define their place in the social world [9]. Realizing a range of intersecting dimensions, such as gender, age, and religion [4], young people build their place concerning spheres of both personal relationships and society beyond interpersonal contexts [9,10].

Against this understanding of identity construction as a fluid process as influenced and informed by sociocultural aspects and othering, in this article, we rely on a series of deliberative discussions carried out with young people from southern European countries; we focused on their notion of belonging to different loci (local, regional, global) (see methodology for details), to problematize the way they spontaneously raise values and display their identities. Our research goal encompasses the acknowledgement of how young people relate to these values and the way it might inform identity building processes. Drawing on the notion of narrative identity [11], we explore the mobilization of values, considering narrative as a notion that reflects the values, interests, and conflicts of the social context in which people live and in which identity develops [2]. The engagement of young people with different values—within associations of belonging to local contexts and Europe at large—is of interest to the extent that it supports the understanding of how they are actively engaged in constructing social (and political) identities. Meanwhile, they are developing narratives that might be flexible, multiple, and certainly related to the contingencies of their social and political environment, which extends beyond their locality or state, or beyond Europe [5].

Due to the nature of our data, a layer related to the European identity is mobilized. The values spontaneously raised by young people in the discussions were engaged with in exchanges that encompassed notions of belonging to Europe in the sense of an enlarged community. The conceptualization of "Europe" in these debates was purposefully left open to capture how young people themselves would relate to their regional standpoint, be it Europe as a region or as the Union. A notion of a European identity has been an issue of continuing challenge with renewed interest since the legitimation of the so-called European Union as a geopolitical block [1,3,6,9,12,13]. Boehnke and Fuss [1] observe, however, that a first reference to European identity as a concept dates back to the 1960s. According to these authors, at that time, the poorly defined notion led to an intense contraposition of the European identity with the national ones.

In tandem with the dilemma of an antagonistic positioning of regional and national identities, Aleknonis [12] argues that the contradictions from within the European Union (or Europe itself) regarding a common European identity arise from the postmodern understanding of the concept. It fosters the idealized perception of a common European identity built above national identities and outside existing conflicts between them, therefore, ignoring a potential multilayered character of attachment [2,14]. For that reason, it is argued that European identity should be understood within affective (attachment to Europe on a territorial level) and cognitive (self-perception as European) dimensions as well [12]. In this sense, "identity is more about a feeling, which is hardly compatible with legal or political definitions" [12] (p. 10) and it certainly has an impact on how values are conceptualized.

With more recent work reallocating the contradiction between the European and national identities, thus acknowledging a non-antagonistic relationship between these two domains [1], there has been a turn to the notion of nested identities [15] as a guiding framework. From this perspective, much aligned with the sense of identity plurality, it is possible to conceptualize the various geo-political identities adopted [1], including those stimulated by place and territory [3]. This can be seen through a lens of complementarity, via possible interchanges deriving from a plural identity display [6], rather than logics of opposition [1,9]. It is an enriching avenue because it allows narratives of Europe to be captured that Scalise [2] has referred to as "multi-level stories, a mixture of values and references coming from the local and national cultural heritages and linked to the European post-national plot" (p. 594). It also makes room for contradiction, recognizing different stories about the same events, therefore, incorporating heterogeneity and diversity [12], including in the dimension of values.

In fact, studies focused on the sense of belonging to the European Union and the building of a related identity suggest that factors such as nationality, gender, and social class [4,9,12,14,16] affect how young people display relationships with the block, including social, political, affective, and cognitive attitudes, which can stem from relationships with significant others [17] or historical events [12]. What is interesting and relevant for the work we present here is the acknowledgment that, within the assembly of the European Union, the 1992 Maastricht Treaty legitimized that every national of a block member would automatically become a citizen of the European Union [3,18], unfolding expectations that these citizens would develop a sense of European identity [9]. In a way, our exploration of how young people display their identities, as organized by a debate on values, is directly associated with how they (differently) embody these same values and correlate them with a European trait.

Acknowledging the importance of the role schools play in mediating young people's identity building [6,19,20], in a final part of the paper, we relate the results of our study to the dimension of citizenship education. We recognize identity building as a process of continuing character [16], which is strongly marked by trends of temporary adhesion in adolescence [3]. Moreover, we understand that the period of schooling corresponds to an important phase regarding the sedimentation of belonging [6,13]. Consequently, the school setting makes up a relevant arena where identities are mobilized while values are formed. Thus, we aim to make a point about how the domain of citizenship education can benefit from acknowledgment of the ways young people display their sense of identity and trends of belonging. For that, we invest in a perspective that considers young people as full citizens rather than citizens in the making [21]. In the remainder of this article, we thoroughly explore the methodological procedures producing the data we discuss, present the study results, and come up with a discussion tying up the domains of youth identity building, the narrative of Europe within it, and the possible ways citizenship education can draw on these aspects with the aim of improving practices.

## 2. Materials and Methods

### 2.1. Data Collection and Participants

This study presents the results of 59 deliberative discussions held in small groups with a total of 378 young people between the ages of 11 and 20 (see Table 1 for full characterization of participants). Data were collected in four southern European countries, namely, Portugal, Spain, Italy, and Cyprus. With the exception of Cyprus, where the discussions took place in 2011, all groups were settled in 2015. These discussions of an open-ended character were extracted from a larger body of data consisting of more than 300 discussions held across 20 European states[1]. Collection was performed by a single researcher in the students' language and, hence, an interpreter was sometimes present in the sessions. In each location, schools were asked to select approximately equal numbers of males and females, within two specified age ranges (11 to 15 and 14 to 20). Selected students were expected to be representative of the ethnic mix of the school. Most of the participants, around 40% of all young people involved in the study, were aged between 14 and 16 years old—only 5% were 13 or younger. Our analysis has not targeted differentiation between age groups because no leads suggested relevant differences in approach. Our cohort is justified under the consideration that the schooling years represent a crucial period concerning values since this is the period when young people internalize societal issues and principles as a process.

The conversations revolved around young people's identification with their locality, country, and Europe, without ever providing a definition of Europe, geographical or otherwise. It is important to highlight that no direct questions about values were made, with related discussion usually emerging from issues raised by the discussants within a mostly unstructured conversation. All deliberative discussions started with a request for participants to present themselves. Then, the facilitator would make a summary on aspects young people raised pointing out similarities and differences and move to a question about

what it meant for them to be of their nationality (the mentioned nationality was always of the country where discussions were held). These identity-related topics naturally fed a conversation on values, especially when students were answering about what it meant for them to be of a specific nationality. All participation was voluntary and informed consent was obtained from both students and their legal guardians. Quotations in this paper are anonymized, indicating only the sex (F—female; M—male) and age of participants.

**Table 1.** Participant profile.

| Feature | Country | Portugal | Spain | Italy | Cyprus | Southern Europe Region |
|---|---|---|---|---|---|---|
| Participants (No. of young people) | | 64 | 122 | 137 | 55 | 378 |
| No. of deliberative discussions | | 11 | 20 | 20 | 8 | 59 |
| Age range | | 14–18 | 11–20 | 12–20 | 12–20 | 11–20 |
| Sex | Male | 36 | 53 | 69 | 14 | 172 |
| | Female | 28 | 69 | 68 | 41 | 206 |

Data collection involved at least two places in each country, addressing settlements of various sizes. Most settlements in the southern European countries were of a small scale (52.4%) and, therefore, had a population of less than 100,000 people. Settlements with over 300,000 people corresponded to 35.2% of the total locations involved while medium-size ones represented 13.8% (see Figure 1).

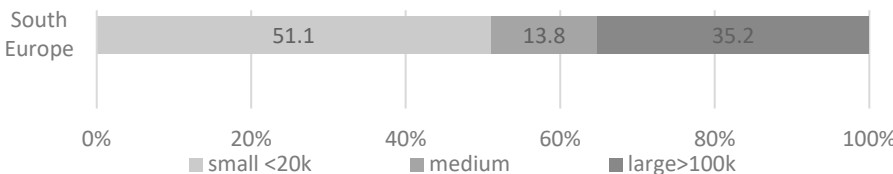

**Figure 1.** Distribution of size of southern European settlements.

Demographic data of a nonconfidential nature were also collected at the time of the deliberative discussions by means of a questionnaire. These data included nationality(ies), citizenship(s), birthplace, home and other languages spoken, and parental occupation and parental origins, in addition to the young person's gender and age. Some of this information will be addressed in the Section 3 where relevant. It is noteworthy to observe that the vast majority of participants' parents were both from the same country where the discussion took place (90.2%). For 5.8% of the young people involved, one or both parents belonged to another European country which is not part of the European Union (EU) or the European Free Trade Association (EFTA); 4.8% had one or both parents from another EU/EFTA country and 3.4% had one or both parents from Africa, Asia, South America, or the Middle East (see Figure 2). This suggests a rather limited diversity in terms of participants' ethnic background.

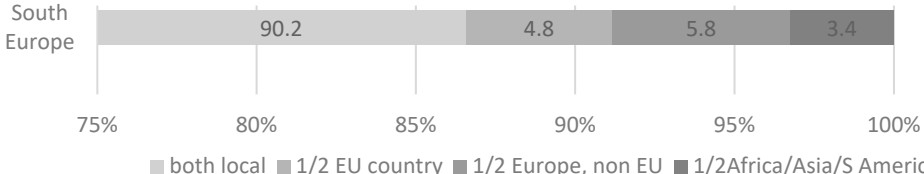

**Figure 2.** Distribution of parental origins in Southern Europe.

### 2.2. On Data Instruments and Analysis

The deliberative discussions are characterized by the fluidity of conversation among participants amidst a few prompts launched by the facilitator [18]. In the study in focus,

the prompts used referred to the sense of identity young people displayed with local, state, regional, and global contexts. The conversation leads were formulated to deal with multiple political identities and, while debating different perceptions, young people ended up referring to a number of values, exemplifying how they attached to them. This means discussion was natural with no structural questions aiming at definitions or understandings of values.

To analyze the values mentioned by the young people participating in the deliberative discussions, the research team[2] established a coding system organized around seven instances of core values: human dignity, freedoms, democracy, equalities of rights, rule of law, human rights in general, and solidarity. Each of these values was targeted either in light of their aspect (types of value in focus) and/or examples mentioned (see Appendix A for values coding framework). Every time an example was given, three characteristics were recorded: location, timing, and othering (i.e., the employment of a third party to illustrate adhesion or refusal of a certain value).

Further analysis highlighted three meta values: (i) Structural values, which encompass the framework that defines the way in which values are created by society; (ii) Core fundamental values, i.e., those defining the general precepts that underpin the various social and political procedures, which, together, fall under the scope of human rights; (iii) Process values, which create legislation, programs, and processes that put the core fundamental values into action[3] (see Table 2). This organization was established and harmonized by the research team after systematic readings of data. We do acknowledge the grouping of values could have been organized in different ways, though. Each value was coded as present or not and was also classified by the aspect and example being used when convenient.

**Table 2.** Structure of the analysis of European values.

| Meta-Value | Value |
| --- | --- |
| Structural values | Democracy<br>Rule of law |
| Core fundamental values | Tolerance of diversity<br>Respect for other cultures<br>Respect for life<br>Safety of others<br>Inclusion in society<br>No capital/harsh punishment<br>Human rights in general |
| Process values | Freedom of movement<br>Fundamental freedoms<br>Equalities<br>Solidarity |

Coding took into account value references made by each individual, registering every occasion upon which a value was mentioned. Therefore, registers capture both the mention of different values and the repetition of the same value by each young person. Additionally, the analysis took into consideration whether when mentioning a value the young person indicated support of approval of it or, on the other hand, they assumed a neutral, ambivalent, or negative posture.

For the purpose of this paper, as well as reporting the reference to the values according to the coding system, we relied on the transcripts to illustrate the landscape of values as addressed by young people. In the Section 3, we problematize the overall mention of values, the most discussed aspects, and the positioning of southern European countries' participants regarding these same values. Next, in the Section 4, we organize the manifestation of values by young people participating in the study by linking their thoughts and expressions to how they display senses of identity and belonging. Finally, we provide

insights about how citizenship education can benefit from such debates to foster practices and ameliorate its agenda.

## 3. Results

This section provides an overview of the values which young people from southern European countries raised during the deliberative discussions and then moves to a more localized, national approach. Obviously, caution should be applied when reading data from a national perspective. The number of participants per country might be low—the reason why our analysis is mostly qualitative, centered on the identification of topics related to the mentioned values.

Because the deliberative discussions were not strictly structured, as explained above, values were only mentioned spontaneously, meaning that not all participants referred to them. This by no means implies lack of interest nor should it be read as a sign of young people's detachment from the theme. On the contrary, this is clearly a natural result of the way groups were conducted methodologically—without direct questions. Overall, 307 young people out of the 378 participants mentioned a value at least once in the discussions. Some of them went further and raised four (*n* = 40), five (*n* = 26), six, or more values (*n* = 26), although the majority referred to a single value (*n* = 83) (see Figure 3).

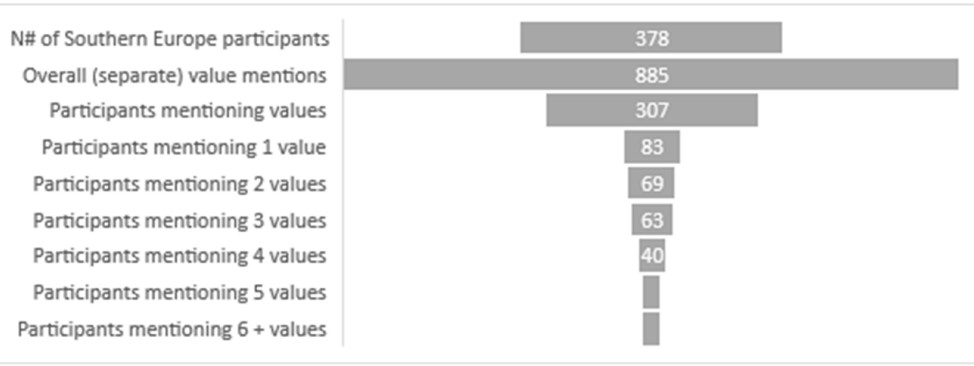

**Figure 3.** Breakdown of mentioned values in Southern Europe.

When considering the mentions within the scope of meta-values, there is a clear predominance of *process values*, with 250 young people referring to them. In this category, the topic of solidarity stands out with 143 participants bringing it into the discussion, which makes this value the one most referred to by different participants. The value of freedom follows closely, with 130 young people mentioning it, either in terms of freedom of movement (*n* = 83) and/or fundamental freedoms (*n* = 57). Next, there is a considerable number of participants referring to *core fundamental values* (*n* = 156), in which the respect for other cultures prevails (*n* = 65), followed by the respect for human life (*n* = 42). Tolerance of diversity and the safety of others are both mentioned by 41 young people. *Structural values* are those raised by the lowest number of participants (*n* = 127), with democracy peaking in this category (*n* = 109) and is actually the second most mentioned value by different people, closely following solidarity. The rule of law, also a structural value, is discussed substantially less, with only 32 people mentioning it (see Figure 4).

### 3.1. Exploring the Landscape of Mentions of Values

In this subsection, we provide detail on the values raised by most young people, shedding light on relevant demographics while also adding context to the mentions. The presentation of results is addressed first by each meta-value grouping, and then a view per country is provided, with a thematic focus on place.

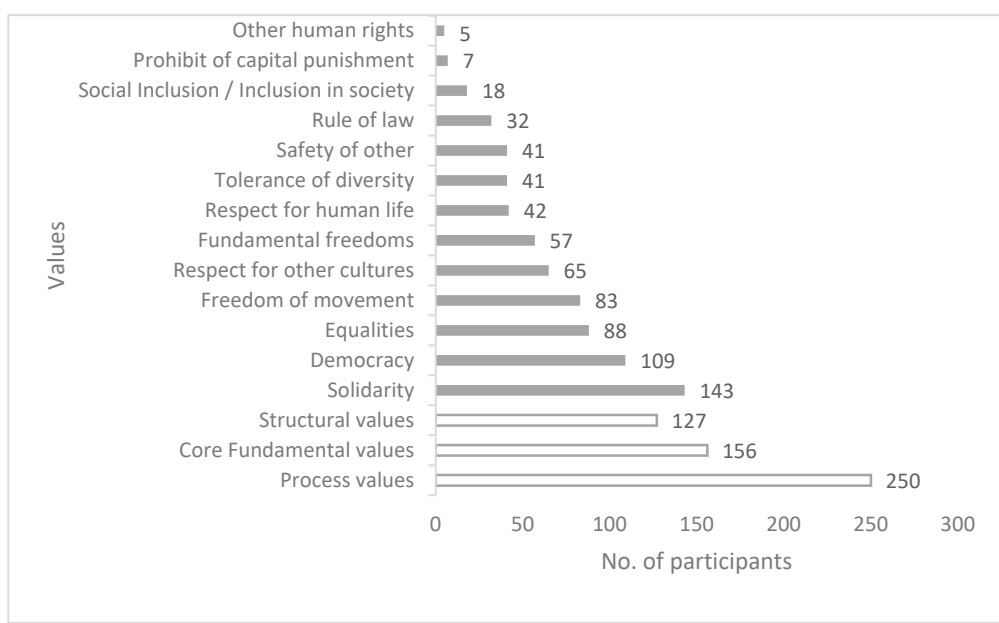

**Figure 4.** Mentions of values by participants.

### 3.1.1. Process Values

The value of solidarity was brought into the discussion by 143 participants, namely 75 girls and 68 boys, and was the most mentioned one in the group of process values. Although data did not suffice to illustrate parental origins by a breakdown of values, a more localized ethnicity can be inferred since 110 young people who mentioned solidarity were monolingual (76.9%)—this trend resembles the overall linguistic ability of participants given that most young people mentioning values speak only one language (see Figure 5). In general, the linguistic ability feature, as expressed throughout this article, suggests that the young people from southern Europe participating in the study have a rather limited background in terms of diversity—be it ethnic or cultural.

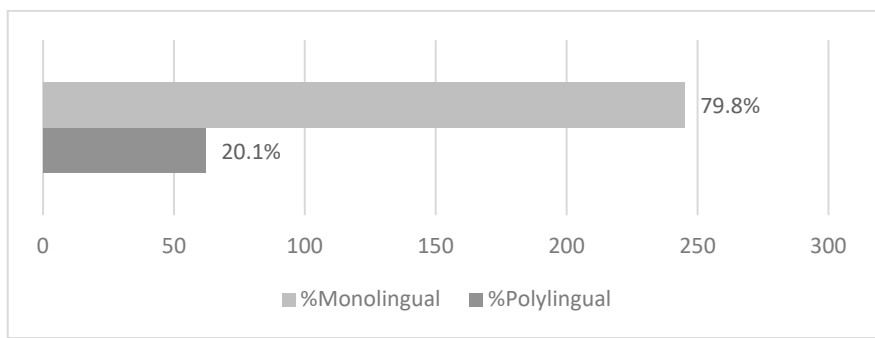

**Figure 5.** Linguistic ability of participants.

The references to solidarity include discussions on issues such as community support, social security, education, and promoting peace. In related examples, young people realize being part of the European Union as enhancing unity and promoting better ways out of collective problems; they discuss the importance of local/national ties while tackling issues and there is recognition that being part of the EU does not automatically resolve contradictions. While a young girl from Cyprus suggests it was a mistake for her country, with a divided political nature, to be integrated into the EU [22], a young Portuguese boy highlights community support by discussing the labor market needs in the follow-up of the late 2000s economic crisis [23,24]. The discussion of such values seems to intertwine a narrative of nested identities [15] for which nationality plays a significant role every time it

disturbs the plot of supranational cohesion. This cohesion is sustained by a strong demand for solidarity ties.

> I think it's important to be proud of the European Union because countries together are more powerful, and can defeat problems together.
>
> (F 13, Spain)
>
> (…) one day Italy will be a good country, and the people will live better than now—but if no one now cares about the problems, then the problems will be bigger, and we'll have nothing.
>
> (F 16, Italy)
>
> (…) I think Europe made a mistake—they accepted Cyprus although we are divided, and they didn't help us re-unite the country.
>
> (F 13, Cyprus)
>
> [I would like us to have] [m]ore money; to have more employment centers.
>
> (M 16, Portugal)

The value of freedom, in turn, was raised by 130 participants. Because it unfolds into two very well-defined categories—freedom of movement and fundamental freedoms—breakdowns were worked separately. Of the 83 young people mentioning freedom of movement, 42 were female, which shows a clear balance in terms of gender. When it comes to the linguistic ability, 75.9% of these participants were monolingual. On the side of young people referring to the value of fundamental freedoms ($n = 57$), most of them were female ($n = 32$), although closely followed by males ($n = 25$). Again, the vast majority of participants were monolingual (85.1%), suggesting no leads in terms of racial/ethnic diversity that could interfere in the way young people reflect upon the value of freedom. The debate around freedom of movement considers different aspects of mobility, covering moving for work/study purposes, leisure, and family reunions. A number of young people acknowledge the possibility of movement between countries as an added advantage, especially when they feel they belong to underprivileged contexts. In this sense, they project identity beyond the national ties, realizing the benefits of belonging to a greater community (see [16]).

> (…) being European is good because we can travel across Europe (…). I'd like to go to other countries, and try to work there—France, or England.
>
> (M 16, Portugal)
>
> I feel more European than Spanish, but I don't travel a lot, far from Spain—but I think if you have a high level of [education] you would have more opportunities for a job outside of Spain (…).
>
> (M 16, Spain)

Young people's references to fundamental freedoms tended to be related to freedom of expression and freedom of speech. Other areas of particular interest for the participants of southern European countries are related to freedom of thought, dress, and religion. In this sense, examples of debate show points of views being sustained by referencing an older person or a different context (region, country, or state), for example. Usually, the selected counterparts are determined as being more conservative within a negative discourse about their behavior. The caption of certain values, such as in the case of freedom of religion, provides a sense of transformation of practices and customs, which is seen as a natural development. This suggests that the participants of the study desire to project a more liberal attitude, assuming their values within a sense of openness that praises diversity. This justifies that, identity-wise, the young people involved in our research feel a need to detach themselves from older generations or contexts they judge as conservative.

> I think older people in general have more closed minds—for example, the old people look at me with my tattoos and go whaw!—why, why?

(F 18, Spain)

[I am pleased about] [f]reedom of speech [in Italy], because in Africa, there isn't this.

(M 13, Italy)

You say young people are not religious—but we are religious—but not so much. Because in the old times, children had to go to the church every day—and now you can do what you want—you are free, but not so free, you have to go only one time if you like (...).

(M 14, Portugal)

Equalities is the *process value* referred to by the least number of participants (*n* = 88), with a balance in terms of gender distribution: 48 boys and 40 girls. Equalities were mostly brought into the discussion in regard to gender. Examples seem to demonstrate a social scenario moving towards a fairer organization, although a few handicaps can still be observed. For that reason, some of the participants take a position of standing for certain movements, such as feminism, claiming to be against the "idea" sustaining the identified barriers to equality. A young girl from Cyprus others older people to make her point about the changes in society not being felt similarly by everyone. Other references to equality contemplate aspects linked to race/ethnicity, socioeconomic status, and LGBT issues.

(...) old people were very religious and stuff like this—and because teenagers now search and have different views, they always complain about us (...). Because at that time women were inferior to men, and because now we are practically the same—it's all so strange for them to see society like this.

(F 15, Cyprus)

(...) I'm a feminist, and it bothers me that a woman receives less than a man at the end of the month, just because she's a woman. I [think] now it's only 13% [less] in my country, but it's not the money, it's the idea.

(F 14, Portugal)

(...) after all, we are leading a good life—considering civil rights, gay people in Italy are not discriminated against. There is democracy. I'm proud of this.

(M 17, Italy)

### 3.1.2. Core Fundamental Values

Regarding core fundamental values, respect for other cultures was the one most mentioned by different young people (*n* = 65). It was brought into focus by 37 boys and 28 girls. Of those participants, 84.6% (*n* = 55) were monolingual, which makes the call to acknowledge other cultures more interest, since this group of participants possibly has rather limited diversity in terms of social background. This value referral portrays a level of controversy, though, with some young people explicitly opposing the inclusion of some ethnic groups. Overall, reactions tend to be positive and grounded in the sense that despite different cultural affiliations, we all belong to the same human nature. Examples focus on ethnic and racial minorities, while migrants and refuges are also targeted. Othering processes suggest changes in mentality over different generations and sometimes reveal a sense of cultural hierarchization, through comparisons that involve local, national, regional, and global spheres. It seems that the participants forge an identity raised on the perspective of the global citizen, within a communitarian framework [13], but sometimes they restrain this characteristic to the places they inhabit.

The refugees aren't a problem—the problem is that some countries that say that they won't take the refugees The refugees come here during the war, and when the war finishes, they go back to Syria.

(M 14, Spain)

> I believe the European Union is more civilized that the rest of the world—and we are not as civilized as the other European nations, based on our everyday behavior.
>
> (F 12, Cyprus)
>
> (. . .) and here, only the Gypsies have free books—and they don't use it, they throw away the books. A friend of mine, he wants to study, he's intelligent, but he doesn't have the money to get books—I don't think that's correct—if they want to give the books, it should be to someone who needs them, not to the Gypsies who don't use them.
>
> (M 15, Portugal)
>
> I think maybe we think a little bit in a different way [compared to older generations], maybe because we are born into a society that is more open to the other cultures, so I think that we have had more contact with other cultures and other countries since we were very, very young. I think this is what makes the difference, we can see how other countries work. (. . .). We should work on our mentality—and it's very hard for an older person to change their mentality, but it's easier for young people.
>
> (F 16, Italy)

Next, the value of respect for human life emerged as the second most mentioned in the group of core fundamental values. It was raised by 23 boys and 19 girls, with a total of 42 mentions. Here, the higher presence of polylingual participants is of interest, who represented 42.8% of the young people mentioning the value. Mentions cover discussions centered especially on the case of refugees or migrants. Participants engage with news, older people's opinions and government practices to demonstrate (mostly) positive feelings towards people from other countries that might be undergoing any type of vulnerability. Like the discourse around respect for other cultures, the viewpoint is that of humans as a big family. Records contradicting an integrative perspective are found, albeit in low numbers.

> I don't think Hungary should take the refugees because Germany tells them to—they should take them because (. . .) they are humans, and they should not take them just because someone tells them to, but because they want to. They should want to take them because of humanity.
>
> (F 16, Spain)
>
> (. . .) when we speak about Europe, yes, we have to speak about a big family, that has to help the different members—and I think sometimes this doesn't happen—for example, immigrants, the majority of immigrants arrive in Italy, and then the other countries in the European Union don't help Italy with these immigrants. (. . .). Maybe if the other countries helped each other, there might be a better situation.
>
> (M 17, Italy)
>
> (. . .) there are a lot of homeless people in Portugal, or who don't have a job, and the refugees are getting a better chance to get a job than the Portuguese people. I think European countries should help people in their own country.
>
> (M 15, Portugal)

The top three of the most mentioned core fundamental values is completed by the topic of safety of the other and tolerance of diversity, each raised by 41 young people. Safety of the other was mentioned by 41 participants, namely, 20 boys and 21 girls. The vast majority of participants who mentioned this value were monolingual, corresponding to 82.9% of the young people referring to it ($n = 34$). Respect for the safety of others regards the protection of the physical integrity and dignity of all individuals and in the case of southern Europe countries; the records are mostly about the extension of these principles to refugees, immigrants, and ethnic or racial minorities. A few quotes related to instances of random violence without specifying a group. Dissonant opinions are captured, with a

few young people expressing their distrust about particular ethnic populations, as in the case of Roma.

> (...) I'm Syrian so I know a lot about these things [refugees]. My parents talk to me about these things. (...). Now I'm in an association of Spanish people and (...) teenagers who are helping (...) Syrian families cross Spain. For example, we wait for them at the station, give them directions on how they can get through Spain and go to Germany or wherever they want to go. We should put our hopes in teenagers—we are the ones that are working for a better life for them, more than the government, or whatever.
>
> (F 16, Spain)
>
> Personally, I don't like Gypsies. I get mad when I see them. I live on a farm, and I have them near me, and they are always doing stuff, stealing. They are (...) normal people, they should work too, but they just don't take the opportunities. They have benefits from the government, but they don't work. They have the age, the skills to work—they just don't want to do that, they just steal and they do stuff like that.
>
> (M 17, Portugal)

Tolerance of diversity, in turn, was referred to by 21 boys and 20 girls, who were mainly monolingual (*n* = 29)—again, suggesting that a possible localized ethnicity heritage does not negatively interfere how young people participating in the study approach plurality. The focal points in the mentions revolved around race and ethnicity issues, with significant discussions being held in terms of religion or sexual diversity as well. Young people tend to be pro diversity, illustrating their opinions in opposition to others who might not hold the same point of view. In the case of Cyprus, the debate encompasses the nature of the country's divided identity, with a strong focus on how the population see each other based on their ethnic origins.

> There is a difference [between Turkish and Cypriots] in the way we speak, for example, we speak differently. There is a difference in the way we are treated—if a person goes to the seaside (...) they are looked at differently. And here in Cyprus there is not that restriction. There are cultural differences—distinctive cultural differences between the two, and each group has its own cultural characteristics.
>
> (F 15, Cyprus)
>
> Here in Spain, we have a lot of racism. We help people in the streets, but sometimes we are very cruel to people from other nations. I know some people that are very cruel to people from other nations, because they have a different skin color, or some people about the Chinese because of their eyes, and I think that's a very big fault in Spanish society (...).
>
> (M 13, Spain)
>
> In Italy, for example, the LGBT community isn't recognized by anyone, and I think it's a problem. If we don't support each other, how can we change the world?
>
> (M 16, Italy)

Other mentions in the group of core fundamental values encompass the topics of inclusion in society (*n* = 18), with debates centered on examples related to ethnicity and racism; the prohibition of capital punishment (*n* = 7), with most references regarding death penalty; and other human rights (*n* = 5) corresponding to generalized referrals to rights as a whole.

### 3.1.3. Structural Values

The structural values group is made up of two topics: democracy and rule of law. For this meta-value, democracy stands out by far, with a total of 109 mentions. As shown before, individually, this is the second-most-mentioned value throughout the deliberative

discussions. Of the young people mentioning it, 57 were boys and 52 girls, depicting a certain balance regarding gender. When democracy was brought into the discussion, most of the time the references covered it in general, i.e., without specifying a particular aspect. Other popular themes in the southern countries included the government acting for all people and opposition to dictatorship. Mentions reveal a considerable positive reaction to this value and debates demonstrate a critical attitude of young people towards governments and politicians' behavior.

> In Portuguese politics, we let people walk all over us for a really long time. We've only had one or two revolutions in all our history, and that was only in the end, after we were really mad at the people who were hurting us (...).
>
> (F 17, Portugal)
>
> (...) I have the same political rights as the others, that's important, because people are [treated] equally (...).
>
> (F 19, Italy)
>
> Because in the old times there was a dictator, and he told you what you had to do, and what you had to say, and if you didn't think that way, I'll go to your house and kill you. Now we are more free, not exactly free, because we are controlled by the government, (...) [but] we can feel like we want, be like we want to be, not like other people want us to be.
>
> (F 17, Spain)
>
> I think they are mixing religion and the state [in Turkey]—that should not be mixed up with the state.
>
> (F 16, Cyprus)

The topic of rule of law, on the other hand, joins the group of individual values infrequently mentioned. It totals 32 mentions, raised by 19 girls and 13 boys. The foci of the discussions concerned mostly the view that laws apply to everyone, with some agreement with the idea that the law and the legal system are of ultimate importance. It seems that a communitarian perspective of citizenship remains, with a view on everyone as equals.

> (...) we are different countries, but we are subject to common rules, that you have to apply to be European. (...) we are organized and we've got common laws.
>
> (M 15, Spain)
>
> I don't think we should say that "We don't respect the rules, so Italy is not an honest country". We can say we don't respect the rules so we can change to be better. I'm very optimistic.
>
> (F 16, Italy)

### 3.2. Exploring the Landscape of Mentions of Values in Each Country

The overview of mentions of values by country is an exercise which is only interesting in terms of qualitatively acknowledging the themes approached in each context. It can support a reflection on citizenship education curricula especially because the values raised through the debates were spontaneously mentioned by the young people participating in them. Nevertheless, as previously reinforced, this analysis is by no means statistically relevant.

In Portugal, a total of 64 young people were involved in 11 deliberative discussions. These groups were conducted in Lisbon, the capital, and Faro, a district capital of medium settlement size, in the south of the country. Of the young people participating, most of them referred to the value of freedom ($n = 27$), with 18 participants focusing on freedom of movement and 11 discussing fundamental freedoms. The second value with most mentions by different participants was equalities, which was brought into debate by 24 young people. The Portuguese top three is completed by the value of solidarity, which was also mentioned by 24 participants.

The mentions of freedom of movement agree with the benefits of circulating around Europe, building an argument that it empowers personal trajectories. Discussions on fundamental freedoms are a bit more diverse, including aspects related to expression/thought, dress, and religion. There is the belief that society is becoming more liberal and accepting of diversity in terms of customs—with most examples centered in the Portuguese context. When debates contemplated equalities, there was a predominance of examples related to gender and LGBT. Young Portuguese people seem to positively reinforce the value of equality, constantly comparing the contemporary generation to the older ones. In their view, again, society is transforming towards a more inclusive and tolerant environment and, as such, equality should be reinforced. Some participants refer to frustration at seeing women's pay being behind in the labor market or LGBT people being excluded from general rights such as adoption. Mentions of solidarity correspond to the general sense of community support within the European Union and the feeling of security it upholds.

> I think that older Portuguese people (…) are more judgmental than younger people. We are OK (…) if a guy wants to wear pink and green and have tattoos, we are totally OK with that—if he wants to, we accept that and we won't judge him. I think older people judge more.
>
> (F 14, Portugal)
>
> It makes me very (…) mad that gay couples can't adopt babies yet, and now we are about to change that [from 2016 on], and gay people could also only donate blood this year [2015]—that also make me very angry. We need to change; we need to grow up.
>
> (F 16, Portugal)
>
> (…) the support of the other countries, in the currency, (…) the trade with the other countries in the European Union, so, some kind of assurance and protection, in case something goes wrong.
>
> (M 18, Portugal)

Consisting of 122 participants, most young Spanish people mentioned the value of solidarity ($n = 57$). Next, democracy is the value most brought up by different people, with 42 individuals raising discussions in this direction. A little behind, freedom ($n = 40$) came into the spotlight with 26 participants approaching the topic of fundamental freedoms while 20 referred to freedom of movement. In Spain, a total of 20 deliberative discussions took place, spread around six settlements in all areas of the country, including the capital city of Madrid.

When referring to solidarity, young people significantly discuss the community support dimension and another theme that stands out refers to social security. Using different states and social groups as a reference, or simply commenting on the European region, young people participating in the study discuss poverty and the impossibility of holding decent standards in housing, for example. The value of democracy was much debated in general terms, aimed at issues of corruption, different government systems (monarchy vs. republicanism) and sometimes making references to the claims on independence from the region of Catalonia. In line with these perspectives, there is a strong positioning in the sense that the government should act for all people. Mentions against the dictatorial period of Spain were also registered. For the case of freedom, the possibilities of moving around are seen as positive since they can maximize better opportunities for life, while referrals to fundamental freedoms focus on issues of expression and thought and there are several comparisons to a time when the country lived under a dictatorship.

> (…) the Spanish people are in a crisis, and can't pay for their houses, and sometimes live in the street, or can't pay for the food for their children. Just because the economy is not so good. (…) we are in crisis, and there are people that haven't got work, jobs, and can't eat, so it's a pity.
>
> (F 13, Spain)

> A lot of people in Catalonia think that Catalonia has to be independent, so they must have a referendum to say what they think—if there are most people that think that, well, Catalonia has to be independent. Because if there are more people that want to leave, they should [a referendum was held the day before this FG took place].
>
> (M 14, Spain)
>
> (…) a safe place is anywhere you can say your opinion or your point of view, without anyone being angry with you.
>
> (M 12, Spain)

In Italy, another 20 discussions were held with the participation of 137 young people, the highest number of the southern European countries. They were conducted in five different locations, both in the north and the south of the country, excluding the capital city of Rome. The value of freedom is the most mentioned by different participants, with 45 young people naming it. Of those, 31 discussed the issue of freedom of movement and 14 engaged with the topic of fundamental freedoms. The next most-represented value was that of democracy with 42 participants mentioning it, the exact same number of young people who raised the topic of solidarity.

In terms of freedom of movement, there are no differences to be observed in comparison to other southern countries. Young Italian people refer to it as a positive aspect, considering the scope of the European Union, and suggest it can enable them to be happier as citizens. When it comes to fundamental freedoms, the topics mainly center around the issues of thought and expression. It is usual that participants refer to other contexts such as Russia or Africa to express their appreciation for having the liberty to think as a common value. More specific mentions refer to freedom of dress and the need to respect everyone's style. In terms of democracy, young people especially debate the fact that the government should act for all people and, as such, they should also be respectful of the legal system, working as models, rather than exceptions. There is great criticism of corruption and some reflection around voting, in the sense that if voting were exercised with responsibility the general political scenario would improve in the country. In turn, solidarity is reflected upon regarding ties to community, with references that, as a block, Europe can support and respond to any serious vulnerabilities. Mentions in terms of peacekeeping are also frequent, which highlights the idea that Europe is united in maintaining its standards.

> We're in the European Union, where we are together. We have freedom of thought.
>
> (F 13, Italy)
>
> My idea is that if people change their minds, automatically the new politicians should change—it's up to the voters.
>
> (M 16, Italy)
>
> (…) when there were the terrorist attacks in France [reference to Charlie Hebdo attacks, which took place the month before the FG], we all felt European.
>
> (M 13, Italy)

The last southern European country under analysis, Cyprus, has the lowest number of participants, with 55 young people distributed into eight deliberative discussions. These groups were organized in four different locations on the island, covering territories of the Republic of Cyprus (the "south") and the Turkish Republic of Northern Cyprus ("the north"), the latter which is not internationally recognized. Most young Cypriot people mentioned the values of solidarity and respect for other cultures, each referred to by 20 participants. Again, it was the topic of freedom that occupied a great number of mentions, with 18 Cypriots referring to it. Of those mentions, 14 focused on the topic of freedom of movement while another six concentrated on the idea of fundamental freedoms.

Discussions in Cyprus were clearly influenced by the sociopolitical divide of the island. As such, conversations about different values tended to partially reflect a viewpoint

concerning this division. Mentions of solidarity reflected the logics of community support, within the understanding of integration—or its absence. Young Cypriots believe the history of the island led its inhabitants to be more supportive of each other, especially within localities. In general, there is controversy about entry into the European Union and the corresponding effects. While some believe it was a successful move, other understand it did not help to improve the nation's union. Within solidarity, and because of the political scenario again, a lot of mentions concerned the issue of peacekeeping. In this sense, the participants' consensus on its importance is notable, although Greek and Turkish Cypriots, to some extent, blame each other for not reaching an integrative agreement in national terms.

The topic of the political divide and the diverse ethnic nature of Cyprus might explain the many references to the value of respecting other cultures. These quotes reaffirm the idea of everyone being human and part of the same family, which justifies treatment without hierarchies. Finally, trends on the value of freedoms are like those observed in the different southern countries. Various young people, especially Turkish Cypriots, refer to Turkey and the possibility of it becoming part of the European Union as a positive act, since it could foster the mobility of those living in the Turkish side of Cyprus. References to fundamental freedoms are quite rare (*n* = 6) compared to freedom of movement and usually contain a dimension of reflection concerning religion and the need to be open about its diversity.

> When we were about to join the EU, I thought that something would change in the situation in Cyprus, but now something is happening about our political problem, but we don't see much happening; they haven't helped us a lot.
>
> (M 15, Cyprus)
>
> I don't think you can separate people because of their views and religion. We are all the same. We need to look at everyone as brothers and sisters, we need to stick together: in the work environment, in the friendship environment, and we should support one another.
>
> (F 14, Cyprus)
>
> I'm not really sure if we get it [Turkey enters the EU], we may have more rights. (. . .). It could be easy for travel into other countries.
>
> (F 15, Cyprus)

## 4. Discussion

The references to values by young people from southern Europe participating in our study and the intersecting contexts framing the mentions help clarifying how sociocultural aspects play an important role in mediating identity building [2,3,5,6,12,18]. In terms of our data, first, it is relevant to recognize the strong dimension of a European identity summoned by the participants when discussing issues that inform the values they spontaneously raised. This identity seems to be constituted in varying forms [2,12], with reflection upon values revealing an interesting movement of othering. It sometimes works to affirm a group participation or adhesion (a race/ethnic nature, a generation, or a nation) and at other times functions as an expression of distancing [3], a tool to distinguish one's positioning relative to a certain topic—usually of controversial nature—beyond the building of multilayered interfaces [5,6].

The high level of expression of *process values* such as solidarity and equalities seem to be good news given that young people relate to it mostly in a positive way, although dispute is detected. In terms of the emergence or stabilization of a European identity, it is noteworthy how participants consider issues such as community support and equalities in general as unquestionably integrating the meaning of Europe, narratives of which obviously navigate through a mixture of values and references [2] while also rearranging national histories [1,12]. Usually, young people's examples referenced different states or older generations that they do not consider to be respectful of these values to reinforce a sense of disconnection from the conception of Europe as they narrate it—either as a

geopolitical block or an abstract community (see [14] for a debate on the multilayered character of attachment). Of course, this representation of the other may not correspond to how these "others" conceptualize values themselves. However, it is relevant since it helps apprehend how young people construct their idea of these groups [4] and then attach their identities to such groupings [7,8], be it states or peoples.

Othering processes put in place while discussing the nature of gender/race/ethnic (in)equality support the comprehension that the same values might be dealt with and approached by peoples in different ways across countries, emphasizing the importance of the locality when framing a possible construction of European identity [1–3,5]. This reinforces the notion of Europe as plural and sheds light on the way local/familiar histories play a part in how the narration of identity and the strength of belonging is mediated [9,12] within a structure of nested identities [15]. The literature has been firm in sustaining that different features such as gender, class, race, etc., inform how citizens connect to the sense of being European [9,14,16]. Our approach to the data does not allow us to be categorical in the specificity of each of these traces, but the discussions held by the young people allow us to apprehend how they combine their own histories (personal, societal, etc.) and belongings (gender, class, etc.) in mobilizing a sense of attachment. A look at the way the value of freedom is mobilized, for example, reveals an instrumental approach concerning Europe that might encompass layers related to class. It appears that participants who recognize more aspects of precarity in their contexts attribute more significance to this value—reaffirming their sense of belonging to a geographical area beyond their national state, while sustaining an identity informed by values of diversity and openness. In that regard, a sense of attachment seems to correspond to the level of opportunity available in each sphere—when acknowledging better opportunities abroad, a young person tends to add value to the aspect of mobility, something possibly maximized according to socioeconomic status (see [2] about social class and identification with Europe).

The way young people discuss core fundamental values provides further evidence of how values are displayed and identity building is influenced by events happening in a familiar environment [4,18]. The references to refugees, migrants, and the contrasting opinions involving different and local ethnic communities all derive from perspectives aligned with facts happening in a time that is current or very recent. This means young people's identity building is informed by social and cultural spheres surrounding them closely, although they can apply layers to these processes derived from practically anywhere nowadays [25]. History also has a say here, especially visible through the mobilization of structural values. The number of references to the issue of democracy and the quality debate over the importance of the government acting for all does not seem to be aleatory. There is clear opposition to dictatorship, especially when young people mention states which they do not consider to be ruled in a democratic way. Southern European countries have an intensive history of dictatorships [26,27], and this might have an influence on the narrative these young people organize about what their contexts are and what they would like them to be politically. This is an important finding in a time when the political interest of young people is suggested to be decreasing [28].

When reflecting upon themes raised by country, the place of history becomes even more visible within the discussions in Cyprus. It is the only country where respect for other cultures emerged among the three most-referenced values by different participants. Knowing the conflicting sociopolitical nature of this state [22,27], it seems only natural that young people's examples orbited around the meaning of their nation's divided nature. Again, from a positive perspective, much of the expression was in favor of a sense of community and unity but not without criticism of a European Union that is seen as incapable of promoting such values beyond discourse—and, therefore, perhaps institutionally responsible for a sense of detached identity.

One of the most interesting points of the debates held in our data collection is that somehow the topic of values was spontaneously raised. The mention of values was not guided or oriented and, therefore, there was authenticity in how young people participating

in the research engaged in the debate. The construction of an atmosphere where everyone could express their thoughts and opinions, without judgment, allowed the expression of a not infrequently negative relationship with values such as equality, respect for other cultures, or even solidarity. The way students freely discussed values seems to be an interesting approach to be adopted in the context of citizenship education curricula as integrated in schools.

Young people display and express citizenship identities that conjugate different conceptual models of what a being a citizen means (liberal, republican, communitarian, etc.) [13], which implies different positioning in how they perceive social life. Teachers, for their part, tend to face controversy as something problematic and avoid dealing with it [3]. It is also known that the place of citizenship education in the curriculum of formal education is itself a field of tension, with disputed views about its role, goals, and strategies [28]. Across Europe, different policies and curricula have been put in place, with strategies varying from defining citizenship education as a compulsory separate subject to a cross-curricular theme. The southern European countries targeted in our study reflect the diversity in policy and curricula approaches. In Portugal, for example, citizenship education is a compulsory separate subject from the fifth to the ninth grade, but it works as a cross-curricular theme in secondary education (tenth to twelfth grade). A similar blended perspective is also dominant in Spain. In the case of Cyprus, compulsory separate subjects on citizenship education have recently been supplanted by integrated approaches, while Italy is going in the opposite direction, i.e., switching an integrated approach for a separate subject model [29].

Independently of the adopted approach, content-wise, we believe teaching strategies would benefit from a participatory perspective in which young people, as the ready-made citizens they are [21], can actually inform the debate on different themes[4], with potential reasoning for controversy. Engaging students in models of learning where they are active and feel safe to express their opinion would stimulate the strengthening of citizenship, as it builds from debate, discussion, and controversy, demanding the ability to deploy skills for arguing and reasoning in conflicting arenas [3]. In a way, this type of dynamic is already going on in schools where a more critical view on citizenship education issues is pursued. As pointed out by Ribeiro and colleagues [28], although the experience of democracy should be learned in context, organized moments in which students can interrogate and reflect on this practice and related dimensions can contribute to the ownership of its respective meanings, promoting a more critical and conscious attitude of young people about society, their role as citizens—and the values they advocate, we would add.

School's socializing role does not play a part outside its students' identity construction, which is affected by different dimension of social life; instead, it is part of it [19,20]. In this sense, we argue that, departing from identity building knowledge, the kind of work to develop in schools on the grounds of citizenship education is not that which impedes the expression of certain values, senses of attachment, and belonging. On the contrary, the investment should be in activities that accommodate different narratives, allowing room to reflect upon one's feelings of belonging and processes of alterities [17] and the way they contribute to becoming who we are as individuals and as a society. It is then all about navigating the different identity narratives together.

**Author Contributions:** Conceptualization, T.F., L.T.D. and F.P.; methodology, T.F., L.T.D. and F.P.; formal analysis, T.F., L.T.D. and F.P.; investigation, T.F., L.T.D. and F.P.; writing—original draft preparation, T.F., L.T.D. and F.P.; review and editing, T.F., L.T.D. and F.P. All authors have read and agreed to the published version of the manuscript.

**Funding:** This research was funded by the Erasmus Jean Monnet Network Citizenship Education in the Context of European Values—The Educational Aspect (621298-EPP-1-2020-1-CZ-EPPJMO-NETWORK). Additionally, this work benefited from the multi-annual funding awarded to CIIE (grants UIDB/00167/2020 and UIDP/00167/2020) and to CeiED under the project doi 1054499/UIDB/4114/2020 by the Portuguese Foundation for Science and Technology, IP (FCT).

**Institutional Review Board Statement:** The study was conducted in accordance with the Declaration of Helsinki, and ethical approval for the data held in this study was given by London Metropolitan University in 2009 and 2014. Written consent was obtained from all participants (young people) and from the parents or guardians of those under 16 years of age, before each discussion took place. This included permission for the discussions to be recorded, transcribed, and then used in a range of unspecified academic publications, including articles and books, based on their anonymized contributions. These statements are held in the archives of the University.

**Informed Consent Statement:** Informed consent was obtained from all subjects involved in the study.

**Data Availability Statement:** Data are unavailable due to privacy and ethical restrictions related to confidentiality.

**Acknowledgments:** We would like to thank all members of Working Group 1—Knowledge and attitudes of young people about civil society, citizenship and European—from the Jean Monnet Network Citizenship Education in the context of European Values (CitEdEV) for allowing us to apply data originally administered in the scope of the referred project. We would like to direct a special word of thanks to Alistair Ross, who collected all data discussed in this article.

**Conflicts of Interest:** The authors declare no conflicts of interest.

## Appendix A

**Table A1.** Values coding framework.

| Value | Aspect | Example |
| --- | --- | --- |
| **Human dignity** | Respect for other cultures<br>Respect for life<br>Respect for the safety of others<br>Prohibition of capital punishment<br>Prohibition of harsh punishment | Migrants<br>Asylum seekers<br>Refugees<br>Race/ethnicity<br>Socioeconomics<br>Gender<br>LGBT<br>Roma<br>Other |
| **Freedoms** | | Freedom of movement (work, study, leisure, and family)<br>Freedom of expression<br>Freedom of speech<br>Freedom of protest<br>Freedom of thought<br>Freedom of dress<br>Freedom of religion<br>Freedom of the media/press |
| **Democracy** | | Democracy in general<br>Free and fair elections<br>The formation of political parties<br>Freedom to run for elective office<br>The separation of political activity from religious beliefs<br>The expectation that the elected government will act for and secure the rights of all inhabitants<br>The prohibition of dictatorship and dictatorial regimes<br>Democracy in general<br>Other |

**Table A1.** *Cont.*

| Value | Aspect | Example |
|---|---|---|
| **Equality of rights** | | Gender<br>Race/ethnicity<br>Sexual orientation/LGBT<br>Socioeconomics<br>Age/ageism<br>Religious belief<br>Disability |
| **Rule of law** | | For laws to be made by an elected body, through a specifically defined public process<br>For laws to be applicable to all people<br>For the judiciary and judges to be independent of political and governmental bodies<br>For the law to be accessible to all<br>Other |
| **Human rights in general** | Tolerance of diversity<br>The right to inclusion in society<br>Tolerance of diversity<br>The right to inclusion in society | Migrants<br>Asylum seekers<br>Refugees<br>Race/ethnicity<br>Socioeconomics<br>Gender<br>LGBT<br>Roma<br>Other |
| **Solidarity** | | Social security<br>Pensions<br>Healthcare<br>Education<br>Accessibility<br>Cultural provision<br>Public transport<br>Community support<br>Workers' rights/unions<br>Promotion of peace<br>People with a disability<br>Environmental protection<br>Food/air/water standards<br>Sustainable development |

## Notes

[1] The data discussed in this paper were derived from the the Jean Monnet Network CitEdEV—Citizenship Education in the Context of European Values. They were gathered by a single researcher before the inception of the network and sustained the activities developed by Working Group 1 (WG1), of which the authors of this paper are part, which focus on the knowledge and attitudes of young people about civil society, citizenship, and European values. Addressing the data produced in the context of WG1 for the purpose of this publication was agreed among all members.

[2] Within WG1 of the Jean Monnet Network CitEdEV, a small working group was assembled to pilot the analysis. Five researchers, including this paper's main author, jointly developed an initial coding frame and tested it. After agreement on consistency, all transcripts of the 324 deliberative discussions were coded, including those targeted here. The definition of values was also targeted in the pilot for harmonization. In this sense, we acknowledge a few quotes could sign more than one value. In such cases, agreement was made to opt for the nuclear meaning of the passage. This process obviously encompasses a subjective dimension of analysis, typical of similar qualitative studies. All of the paper's authors participated in the analysis procedures. In all, each member analyzed around 30 transcripts.

[3] A full reasoning for values classification is available in the full report made by WG1, under the coordination of Tom Loughran and Alistair Ross [Young People's Understanding of European Values: Enhancing abilities, supporting participation and voice. Report of the Jean Monnet Network Project: Citizenship Education in the Context of European Values (CitEdEV)] (to be published soon). In this document, it is also possible to reach further detail on the context of different values mention by young people and respective analysis.

[4] Although some specificites can be found in the citizenship education curricula from the different southern European countries, in general, the explored themes converge into core areas like human rights, democracy, and sustainability [29].

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
