# Peer review of "Young People’s Construction of Identity in the Context of Southern Europe: Finding Leads for Citizenship Education"

_societies, doi:10.3390/soc14010009_

Round 1

Reviewer 1 Report

Comments and Suggestions for Authors

Thank you for the opportunity to read the article. I believe that the research findings presented there may be of interest to many scholars studying youth identities in the European context. However, I have a few comments that might improve the academic quality of the article. I present them below:

1)      In my opinion it is necessary to clearly indicate the purpose of the research in the 'introduction' section;

2)      Besides, it is necessary to complete the information on how the study participants were selected;

3)      There is a large age difference between the participants in the study (between 11-20); I the article there is only the information about the age range of the participants in each country; I think that an extension is needed. It is quite important to indicate the detailed age structure of the research sample as the ages of the respondents were not similar. Did age differentiate the respondents' answers? Did the researchers analyse the utterances taking into account the age of the respondents? This requires comment.

4)      It is not clear to me for what reason the researchers present the results in relation to the linguistic ability of participants. Clarification is needed in this regard.

5)      Please, consider changing the title of the article. In my opinion, this is not an article about education for citizenship in Southern Europe. Besides, the 'disscusion' section is also something to think about. The content of this section makes sense of course, however it does not follow from the results of the research presented earlier. In my opinion, education is a bit 'tacked on' here.

6)      I also would request more precision in the language. The article shows that the authors are aware of the limitations of their own research, at the same time they generalise in many places by writing about ‘youth from Southern Europe’, while at most they can speak about the sampled youth from selected Southern European countries. I think this is a mistake that can be easily corrected.

Author Response

Please see atachment.

Reviewer 2 Report

Comments and Suggestions for Authors

Review: Educating for Citizenship in Southern Europe: What Can We Learn from Young People’s Construction of Identity

This makes a highly valuable contribution to understanding young peoples’ views on a range of issues of relevance to democratic societies. The methodological approach is sound and the data collection is clearly rigorous, encompassing an impressing range of data. The data is presented in a clear and interesting way, with the use of graphs helping the reader to gain a broader perspective on the discussions.

In terms of the methodology, it is clear that these were unstructured, deliberative discussions, which is key to the generation of such rich data. Given that the connection between values and identity is at the heart of the analysis, it would be good to know how the discussions were framed. What opening question were young people asked? i.e. did this discussion of values emerge in the context of a discussion about their identity? I assume this is the case, but this is not clear.

The connection between the focus on values and young people’s senses of identity and belonging is interesting and original. This focus and the connection between the two could be made more explicit in the abstract and introduction. The notion of narrative identity is central in making the connection between the young peoples’ discussion of values and their sense of identity. This connection could also be made more explicit in the presentation of the findings. As it is, discussion of identity is left to the ‘discussion’ section, where it is covered quite concisely. It would be good to see the story more obviously throughout – this may be because the analysis hangs so heavily on the values themselves, so there is explicit little commentary on how these values relate to identity. The contribution of this analysis to discussions on identity formation would be strengthened by a clearer ‘identity’ thread throughout the findings section.

The grouping of values into the three types of meta-value is helpful in terms of making sense of the data and in providing a coherence to the analysis of what matters to these young people. Arguably, the ‘process values’ could equally be read as ‘core values’, For example, freedom of movement could be seen a s fundamental right. Given that this process is necessarily subjective, the grouping would be strengthened by acknowledgment of this and the fact that these could be interpreted otherwise. This would acknowledge the researchers’ subjectivity. The same could be said in relation to the author’s interpretation and selection of quotes. Of course, much of the data is open to multiple interpretations. For example, the following quote (L645 – 647), used to illustrate ‘freedom’, could, arguably, be better suited to illustrate solidarity:

“I don’t think you can separate people because of their views and religion. We are all the same. We need to look at everyone as brothers and sisters, we need to stick together: in the work environment, in the friendship environment, and we should support one another. (F 14, Cyprus)”.

The analysis would have more rigour and credibility if the subjectivity of the interpretation process was discussed.

With respect to the qualitative data and quotes, it is clear that there is a vast amount of valuable data that the authors have condensed into this article. At times however, the reader is left wanting more – both in terms of quotes that represent the values highlighted and in terms of more in-depth analysis of the quotes that are used. For example, the values of safety and respect for diversity are illustrated in respect to helping refugees and discrimination, which are very interesting, but it would be good to hear how else young people talk about these themes. In particular, as commented above – how does this relate to their understanding of identity? I appreciate that there is a word limit and the authors want to convey the meta story in the data. Perhaps a note to signal that the data presented is only the tip of the iceberg and that further analysis on these themes is forthcoming or can be found elsewhere?

-          A couple of other points on the presentation and discussion of the data:

L275: The quote (participant M16 Portugal) seems out of place here. Is this about solidarity? If so, its needs some explanation.

L304: Religion is highlighted as a value. Does the author mean ‘freedom of religion’? This should be made clearer. The value of freedom of religion is alluded to again in relation to the state and democracy. It is interesting that the young people seem to see religion as something outdated or at least, the way in which older people live their religion (in terms of strict church attendance). This would be an interesting dimension for further focus in terms of their identity.

In several places, the linguistic breakdown of respondents is highlighted. However, the significance of this is not explained or commented upon. Is this linked to how well the respondents were able to convey their views or does it have another significance?

On the discussion section: The links between the discussions and citizenship education needs to be made clearer. The fact that discussion of values emerged naturally is attributed to the existing citizenship education: “Taking this scenario into account, we understand that citizenship education plays a crucial role in the educational scenario. “(L728). Likewise, the exploration of values mentioned by country is said to reflect the citizenship curricula in those countries. Whilst this may be the case, it would be good to hear more about what those curricular look like. How far does citizenship ed in these countries allow for debate on such issues? Is it more content or process focused, for example?  Also, this should be qualified by an acknowledgment that young people are socialized into a set of values by a range of agents (family, media etc).

The paragraph in the discussion that states the authors’ views on the need for more dialogical citizenship education seems to be a key argument. Indeed, the data shows the nuanced ways in which young people understand citizenship and that the kind of deliberative discussion engaged in, enabled them to explore and express the complexity of their identity and belonging. I think this could be explored further in order to evidence the need for such approaches within the curriculum. This could be related to existing approaches within critical citizenship education that foreground deliberative discussion and exploration of identity.

-        

I hope that this is helpful and that the authors revise this and resubmit. The evidencing of the multifaceted nature of identity and its relationship to social, political and historical contexts is of profound importance to understanding and living well together. Evidencing this and its relevance for citizenship education is therefore an urgent need.

Comments on the Quality of English Language

  In places, more clarity in needed in word choice:

P1L28: ‘displays’ does not seem an obvious word choice here. Would ‘plays’ work better?

P1, L30: Othering is explained as “i.e., the individuals mobilize their repertoire of identities according to who they are interacting with”. There is another meaning of ‘othering’ in identity studies, whereby the majority marginalise the minority through focusing on their difference to the assumed norm. What the authors are describing here seems to be more about the social contingency of identity. Othering, to me anyway, signals an acknowledgment of a power differential.

L300: “In this sense, examples of debate show point of views being sustained though othering an older person or a different context (region, country, state), for example.” Again, the use of the term ‘othering’ is not clear here.

-          There are a few instances where very long sentences would benefit from being broken down or rephrased so that the meaning is clearer:

L63-65. This sentence is a bit too wordy and could be usefully broken down.

L706 – 709: “The number of references to the issue of democracy and the quality debate over the importance of government acting for all does not seem to be aleatory, while there is clear opposition to dictatorship, at many times othered in the figure of states which young people do not recognize as respectful of democratic rules”.

P3, L111-114: This sentence would benefit from some unpacking. It seems there is too much here that has been summarised so succinctly that the contribution of the studies quoted is not expressed sufficiently. It would be good to know a little more about these aspects of identity development in schooling. This would also provide a more solid base for the later argument of the study’s relevance to citizenship education.

L634: The following sentence needs rephrasing to make the meaning clear. “Numerous quotes, mainly from Turkish Cypriots, other Turkey and the possibility of it becoming part of the European Union, within a positive act, since it could foster the mobility of those living in the Turkish side of Cyprus.”

The following typos need attention:

P2, L49: ‘construct’ needs ‘ion’ adding

L54 – additional brackets

L90: Typo in ‘identities’.

L134: Do you mean students’ ‘identification’ with?

L232: Needs a ‘s’ added to ‘other’.

L301: Typos on “show point of views”

L303: ‘positively’ should read ‘positive’. Here, religion is referred to as a value. Does the author mean freedom of religion. This is an interesting area for consideration in relation to identity.

Round 2

Reviewer 1 Report

Comments and Suggestions for Authors

I consider the revisions made by the authors sufficient for the article to be accepted.